# CAGGLE: COLOR-AWARE GUIDANCE WITH GLOBAL AND LOCAL PROMPTS FOR EXPOSURE CORRECTION

## ABSTRACT

In real-world exposure correction, achieving high-quality images requires addressing multi-exposure conditions and managing images containing locally varying brightness. While recent deep learning models have improved image correction across various exposure levels, they often struggle in complex scenarios where both under- and over-exposure coexist within an image or in over-saturated areas with sparse pixel information. In this paper, we tackle these challenges by proposing a color-aware guidance that employs a global prompt for tone adjustment and a local prompt for maintaining color consistency of the output. To achieve this, we present a novel Prompt Interaction Module (PIM) that seamlessly integrates the global and local prompts with the input image features. Extensive experiments on multi-exposure benchmark datasets demonstrate that our method achieves state-of-the-art performance, outperforming existing exposure correction methods. Our approach sets a new standard in exposure correction, leveraging prompt-based learning for improved color and exposure adjustments.

## 1 INTRODUCTION

In real-world photography, images are captured under various exposure conditions, and incorrect exposure can obscure important details in images. To address this issue, modern cameras offer exposure compensation features, and furthermore, many software-based solutions have been developed to automatically resolve this problem. Despite significant advancements in both hardware and software that have greatly improved image quality and alleviated problems associated with severe under-exposure and over-exposure, challenges remain that often necessitate expert intervention to adjust settings such as aperture, shutter speed, or lighting to adapt to complex environments. While manual adjustments can improve image quality, they are impractical for CCTV systems installed in hard-to-reach areas or for vision cameras in automated factories, which require automated solutions for reliable performance without manual input.

To address this issue, numerous exposure correction methods have been developed. Early research primarily concentrated on addressing either under-exposure or over-exposure, which limited the effectiveness of these methods in managing complex exposure conditions. Considering the limitations of conventional approaches in addressing various exposure issues, recent methods have focused on developing models that can perform multi-exposure correction using a single deep neural network by jointly training on both under- and over-exposure datasets. Among them, MSEC (Afifi et al., 2021) presents a multi-exposure dataset for exposure correction and highlights that addressing multi-exposure issues requires resolving the complex interaction between brightness and structural information in images. Consequently, exposure correction methods have evolved to address the complex degradation of lightness and structure, with approaches such as the use of Fourier transformation (Huang et al., 2022b) and local color distributions (Wang et al., 2022; Li et al., 2024) proposed for modeling this combined degradation. From another perspective, the emphasis on feature-level enhancement is increasing in recent studies. For instance, DA (Wang et al., 2023b) and ERL (Huang et al., 2023) introduce pluggable modules that enhance diverse exposure inputs in the high-level feature space, thereby strengthening existing correction methods.

Although existing studies show significant performance improvements, sparse pixel information from under- and over-exposure can result in inadequate enhancement outcomes. For instance, as shown in Fig. 1, conventional approaches struggle in challenging scenarios where the captured im-

Figure 1: Comparison of exposure correction in dynamic scenes with existing methods. **(Top)** In images with varying exposure issues, our method produces higher-quality results compared to existing approaches. **(Bottom)** CAGGLE also demonstrates comparable reconstruction in cases involving over-saturated regions.

ages are either extremely under-exposed or over-exposed, failing in one or both conditions. Particularly, conventional methods tend to cause color distortion when the captured images include areas that are over-exposed due to a light source. Furthermore, even for the same object, color can be represented differently depending on spatial brightness variations, making it crucial to understand the local color distribution of the input image to achieve accurate enhancement.

Therefore, we propose a novel color-aware network utilizing Local and Global Prompts to capture input-specific local color distribution and achieve natural exposure correction. In this approach, the Local Prompt focuses on correcting spatial and localized features, while the Global Prompt manages overall tone and exposure adjustment. These two learnable prompts interact dynamically within the exposure correction network, offering essential guidance for accurate color and exposure correction. Our color-aware approach for exposure correction is inspired by existing techniques that consider local color distribution for other image enhancement tasks such as LCDPNet (Wang et al., 2022) and CSEC (Li et al., 2024). However, unlike previous approaches, this work introduces Local Prompt that is guided toward linguistically defined color categories in the low-dimensional space, based on the Color Naming model (Van De Weijer et al., 2009). This allows Local Prompt to facilitate robust spatial and color-specific enhancements across varying exposure conditions. As a result, even if the color of the same object varies due to different exposure levels, the correct color can still be accurately restored after applying exposure correction, even in challenging scenarios.

We call this approach **CAGGLE** (**C**olor-**A**ware **G**uidance with **G**lobal and **L**ocal Prompts for **E**xposure Correction). To the best of our knowledge, CAGGLE is the first approach to employ prompt-based learning for exposure correction, establishing a new benchmark in the field. Our main contributions can be summarized as follows:

- CAGGLE leverages the synergy between Global and Local Prompt to enhance image features, demonstrating its effectiveness in correcting the overall image tone and addressing challenges arising from exposure variations as depicted in Fig. 1.

- By integrating the Color Naming model to guide prompt learning, we leverage local color statistics of input to enable color-aware exposure correction, ensuring color consistency.

- CAGGLE outperforms conventional exposure correction methods on multi-exposure datasets, including MSEC (Afifi et al., 2021), SICE (Cai et al., 2018), and LCDP (Wang et al., 2022), achieving state-of-the-art (SOTA) results across all datasets.

## 2 RELATED WORKS

### 2.1 EXPOSURE CORRECTION

Before the advent of deep learning methods in computer vision, exposure correction primarily relied on conventional techniques aimed at improving image contrast. Methods such as Histogram Equalization (Gonzales & Wintz, 1987) and Gamma Correction were widely used to enhance contrast. Additionally, Retinex theory (Land, 1977; Jobson et al., 1997; Rahman et al., 2004) was employed to address not only image contrast but also color constancy problems. Since the introduction of deep neural networks (DNNs), significant progress has been made in various computer vision tasks. In

particular, in the field of exposure correction, DNN-based approaches employing a variety of concepts have emerged. Methods such as (Wang et al., 2019; Wei et al., 2018; Wu et al., 2022; Zhang et al., 2021; 2019), which are based on Retinex theory that decomposes images into reflectance and luminance, have been proposed for under-exposed image enhancement. Additionally, Retinex-based methods like CMEC (Nsampi et al., 2021) have addressed multi-exposure correction through attention mechanisms, while LCDPNet (Wang et al., 2022) introduced an approach that considers local color distribution. Moreover, to address the multi-exposure correction problem, the MSEC (Afifi et al., 2021) and SICE (Cai et al., 2018) datasets were developed for both training and evaluation, with MSEC also proposing a Laplacian pyramid architecture capable of managing multiple exposure conditions. Similarly, ECLNet (Huang et al., 2022c) employed a bilateral activation mechanism to differentiate the treatment of multi-exposure images, while FECNet (Huang et al., 2022b) introduces a lightweight spatial-frequency interaction model based on a Fourier-based approach.

Recent works have increasingly focused on regularizing and enhancing feature maps to improve performance. ENC (Huang et al., 2022a) advanced this effort by incorporating an exposure normalization module that maps varying exposure features to an exposure-invariant feature space. CSNorm (Yao et al., 2023) enhanced the generalization capability of existing methods by selectively normalizing lightness-relevant channels. ERL (Huang et al., 2023) applied regularization for multi-exposure correction, while DA (Wang et al., 2023b) introduced a contrast and detail-aware unit that can be integrated into existing architectures. The latest work, CSEC (Li et al., 2024), modeled color distribution shifts at the feature level.

## 2.2 Prompt Learning

Recently, prompt-based learning methods have gained widespread use in natural language processing for fine-tuning inputs tailored to specific tasks. The process of finding appropriate input prompts was introduced in (Brown et al., 2020). Unlike approaches that focus on finding fixed-format prompts, CoOp (Zhou et al., 2022) introduced a method that treats prompt context as learnable parameters, outperforming handcrafted prompts and demonstrating the superiority of learnable prompts. Following CoOp, several methods have been proposed to generate appropriate prompts (Smith et al., 2023; Derakhshani et al., 2023; He et al., 2022). CODA (Smith et al., 2023) creates input-specific prompts through input-conditioned weights, HyperPrompt (He et al., 2022) addresses multi-task learning with a prompt generator, and bayesian prompt-learning (Derakhshani et al., 2023) models input prompts from a Bayesian perspective, adopting a probabilistic approach. In computer vision, visual prompts refer to methods that modify inputs by applying additional trainable parameters to effectively tune the model. VPT (Jia et al., 2022) shows significant performance improvements over other fine-tuning methods by keeping the large transformer model backbone fixed and applying a small number of trainable visual prompts. In addition, Bahng et al. (2022) proposed visual prompts that can be combined with input images which are effective for CLIP. For low-level vision tasks, PromptIR (Potlapalli et al., 2023) and PromptRestorer (Wang et al., 2023a) use prompts to encode degradation-specific information and guide the restoration network to generalize to different degradation types and levels. In contrast, we propose input-specific global and local prompts for multi-exposure correction, exploring both local and global information.

## 2.3 Color Naming model

Precise color naming ensures consistency across various tasks, enabling reliable image analysis, object recognition, and visual understanding, which are critical for tasks like image annotation, vision research, and photography. *Basic Color Terms* (Berlin & Kay, 1991) identifies semantic universals in the color vocabulary across linguistic boundaries. They show that, despite variations in the number of basic color terms across languages, there is a universal inventory of exactly 11 basic color categories: *black, blue, brown, gray, green, orange, pink, purple, red, white,* and *yellow*. Building on this foundation, a color decomposition model was proposed (Van De Weijer et al., 2009), and this model outputs probability values for each pixel, indicating the likelihood that the pixel belongs to one of the 11 predefined color names based on its sRGB values (Appendix. Fig. 8 (a)). Recent study, Serrano-Lozano et al. (2024) cluster colors with similar hues into 6 broader names (*red, green, blue, orange-brown-yellow, pink-purple,* and *white-gray-black*) to facilitate easy computations (Appendix. Fig. 8 (b)). Our CAGGLE employs these 6 color names to guide the color-aware prompts.

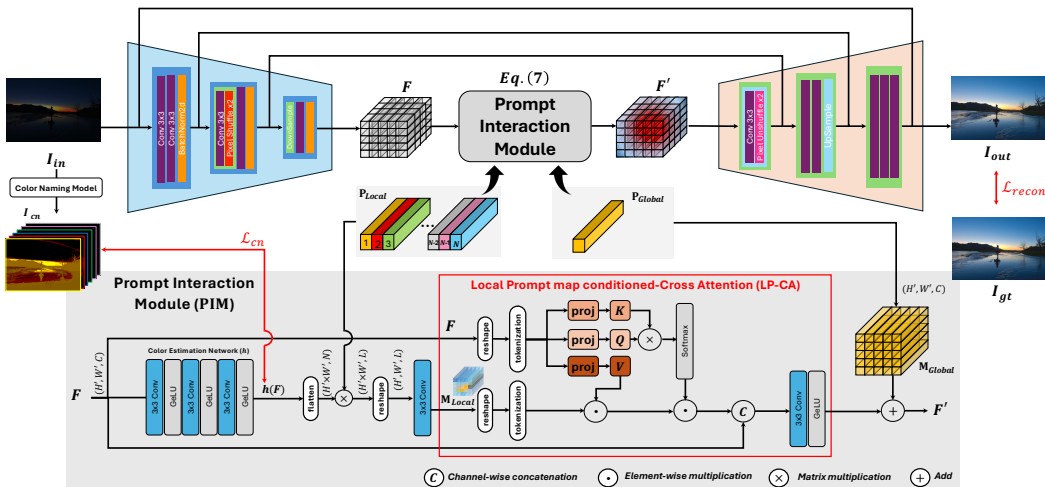

Figure 2: Overall architecture of CAGGLE. CAGGLE follows a simple U-shaped residual network design, with the Prompt Interaction Module (PIM) located between the Encoder and Decoder. Within the PIM, learnable prompts $\mathbf{P}_{Local}$ and $\mathbf{P}_{Global}$ are transformed into $\mathbf{M}_{Local}$ and $\mathbf{M}_{Global}$. The PIM takes $F$ as input and dynamically generates the enhanced feature $F'$ through its interactions with $\mathbf{M}_{Local}$ and $\mathbf{M}_{Global}$.

## 3  PROPOSED METHOD: CAGGLE

### 3.1  OVERALL PIPELINE

Fig. 2 presents the overall architecture of CAGGLE, which consists of three main components: the Encoder, the Decoder, and the Prompt Interaction Module (PIM). CAGGLE is built upon a U-shaped residual structure, where the Encoder processes the poorly exposed input image $I_{in}$ and progressively transforms it into a deep feature map $F$. The PIM serves as a crucial bridge between the Encoder and Decoder, further refining the feature map by leveraging input-specific prompts that dynamically adjust the overall tone and enhance finer details such as color accuracy, contrast, and sharpness of edges. In the final stage, the Decoder takes the enhanced feature representation from the PIM and reconstructs it into a well-exposed output image $I_{out}$, effectively addressing challenging exposure issues.

### 3.2  ENCODER AND DECODER

The Encoder and Decoder are organized into conventional stages, with each stage consisting of multiple convolutional layers that are responsible for progressively transforming the input and output features. In the Encoder, pixel-shuffled downsampling is used to gradually reduce the spatial resolution and batch normalization is used to normalize features, enabling the model to capture abstract and compressed representations from images with varying exposure levels. Conversely, in the Decoder, pixel-shuffled upsampling is applied to restore the spatial resolution of the feature maps, progressively reconstructing the well-exposed output image. Additionally, at each corresponding stage, concatenation operations are applied from the Encoder to the Decoder, facilitating the flow of essential low- and mid-level features from the encoding process into the decoding process, thereby preserving important details and aiding in the final image reconstruction.

For more detailed information, we provide detailed specifications of the Encoder and Decoder architectures in Appendix. A.1.

### 3.3  PROMPT INTERACTION MODULE (PIM)

The purpose of the PIM is to effectively enhance the feature map $F$ extracted from the Encoder through interaction with two learnable prompts: the Local Prompt and Global Prompt. In particular,

to provide appropriate input-specific guidance for Local Prompt, we introduce a color estimation network $h$ which captures spatially varying color information of the input image. The input-specific Local Prompt is further processed through a specially designed cross-attention mechanism, and the combined result with the Global Prompt produces an enhanced feature map, which is then passed to the Decoder.

### 3.3.1 LOCAL PROMPT

Our Local Prompt $\mathbf{P}_{Local}$ consists of $N$ learnable prompt vectors, represented as $\mathbf{P}_{Local} = [\mathbf{P}_{Local}^1, \mathbf{P}_{Local}^2, \ldots, \mathbf{P}_{Local}^N]$, where the i-th local prompt vector $\mathbf{P}_{Local}^i$ is an $C$-dimensional vector. Notably, unlike previous methods (Potlapalli et al., 2023; Jia et al., 2022) that apply prompts globally, differently weighted versions of the Local Prompt are applied at each spatial location, allowing $F$ to be handled in a distinct, localized manner.

Specifically, to provide input-specific local information to the Local Prompt, we combine the result from a shallow color estimation network $h$ with our Local Prompt. Specifically, $h$ takes the feature map $F \in \mathbb{R}^{H' \times W' \times C}$ as input, and outputs a weight map in $\mathbb{R}^{H' \times W' \times N}$ after a softmax operation. This weight map is then flattened to a dimension of $\mathbb{R}^{H'W' \times N}$, and is used to control the importance of the Local Prompt at every spatial location through matrix multiplication. After applying reshaping and a single convolutional layer, it yields the Local Prompt Map as follows:

$$\mathbf{M}_{Local} = \mathbf{Conv_{3 \times 3}}\Big(\mathbf{reshape}(\mathbf{flatten}(h(F)) \times \mathbf{P}_{Local})\Big), \tag{1}$$

where $\mathbf{Conv_{3 \times 3}}$ denotes a $3 \times 3$ convolution layer, and $\mathbf{M}_{Local} \in \mathbb{R}^{H' \times W' \times C}$ represents the Local Prompt Map, containing spatially varying and input-specific information. Notably, by representing our Local Prompt $P_{Local}$ as a set of $N$ vectors where each vector is in $\mathbb{R}^{1 \times C}$, we easily alleviate the issue present in previous prompting methods that required the input image size to match the prompt, a constraint that is often impractical. This approach allows the Local Prompt to be independent of the input image size, offering greater scalability and enabling more effective local prompting.

Moreover, to address the color distortion issues commonly seen in conventional approaches, as in Fig. 1, and to ensure consistent colors across spatially and exposure-varying objects, we design our Local Prompt to capture spatially different color distribution using the Color Naming model and propose a dedicated cross-attention mechanism to effectively leverage the color information.

**Color-Aware Guidance with the Color Naming Model**   To address the existing issue of color distortion, it is crucial that our color estimation network $h$ accurately captures locally varying color information in poorly exposed input images. While $h$ can be trained in an unsupervised manner, we leverage a color naming approach (Serrano-Lozano et al., 2024) and predefined color names. Since color names represent colors in a low-dimensional space, they are well-suited for representing distorted colors in over-exposed or under-exposed input images. This supervision ensures that Local Prompts are weighted to align with the color names, which are more robust to color distortion at each spatial location. Thus, $h$ effectively captures the spatial and local color distributions of the input image, allowing CAGGLE to produce color-consistent outputs regardless of input exposure level. Moreover, this enables $h$ to perform a comprehensive analysis of local structural features, such as edges, in addition to color distribution, ultimately enhancing the quality of $\mathbf{P}_{Local}$.

In this work, we utilize the Color Naming model from Serrano-Lozano et al., which further clusters 11 predefined color names based on sRGB values (Van De Weijer et al., 2009) into 6 categories of similar hues. These 6 categories, grouped by colors that differ only in intensity and share similar hues, can guide the Local Prompt and provide solid guidance for exposure correction. We present a detailed explanation of the loss function used to train $h$ in Sec. 3.4. Notably, as demonstrated in our ablation study in Sec. 4.5, even without incorporating the Color Naming model into the prompt design, CAGGLE exhibits outstanding effectiveness compared to existing methods. However, using the low-dimensional color names as supervision for prompt learning significantly boosts the performance of CAGGLE.

**Local Prompt Map Conditioned Cross-Attention**   The key role of our local prompting is to enhance the feature map $F$ by interacting with the input-specific Local Prompt. To facilitate this

interaction, we propose a novel cross-attention mechanism, **L**ocal **P**rompt map conditioned **C**ross-**A**ttention (**LP-CA**), which can capture long-range dependencies and relationships between distant spatial locations in $F$ and $\mathbf{M}_{Local}$.

First, as illustrated in Fig. 2, $F \in \mathbb{R}^{H' \times W' \times C}$ is reshaped and tokenized with $K$ heads as:

$$X = [X_1, X_2, X_3, \ldots, X_K], \tag{2}$$

where the i-th head $X_i$ is in $\mathbb{R}^{H'W' \times \frac{C}{K}}$, and each head $X_i$ is further projected into key $\mathbf{K}_i \in \mathbb{R}^{H'W' \times \frac{C}{K}}$, query $\mathbf{Q}_i \in \mathbb{R}^{H'W' \times \frac{C}{K}}$, and value $\mathbf{V}_i \in \mathbb{R}^{H'W' \times \frac{C}{K}}$, respectively, as follows:

$$\mathbf{K}_i = X_i W_{K_i}^T, \quad \mathbf{Q}_i = X_i W_{Q_i}^T, \quad \mathbf{V}_i = X_i W_{V_i}^T, \tag{3}$$

where, $W_{K_i}$, $W_{Q_i}$ and $W_{V_i}$ in $\mathbb{R}^{\frac{C}{K} \times \frac{C}{K}}$ represent the learnable parameters of the fully connected layers and $T$ denotes the transpose operation of the matrix. Similarly, we tokenize $\mathbf{M}_{Local}$ and split it into $K$ heads as:

$$Y = [Y_1, Y_2, Y_3, \ldots, Y_K], \tag{4}$$

where $Y_i \in \mathbb{R}^{H'W' \times \frac{C}{K}}$. Then, our cross-attention mechanism employs $Y_i$, as conditional information, to correlate prompt information for $V_i$, represented as:

$$\mathbf{Attn}(F, \mathbf{M}_{Local}) = softmax(\frac{\mathbf{Q}_i \mathbf{K}_i^T}{\sqrt{d}}) \cdot (\mathbf{V}_i \cdot Y_i). \tag{5}$$

$\mathbf{Attn}(F, \mathbf{M}_{Local})$ denotes the output of cross-attention, and $d$ is a learnable parameter that adaptively scales matrix multiplication. Next, $\mathbf{Attn}(F, \mathbf{M}_{Local})$ is concatenated with $F$, followed by a convolution operation and GeLU activation function, producing the **LP-CA** output as:

$$\mathbf{LP\text{-}CA}(F, \mathbf{M}_{Local}) = \mathbf{GeLU}\big(\mathbf{Conv_{3 \times 3}}([F, \mathbf{Attn}(F, \mathbf{M}_{Local})])\big). \tag{6}$$

### 3.3.2 GLOBAL PROMPT

Inspired by previous prompt-based approaches (Zhou et al., 2022; Jia et al., 2022; He et al., 2022), we introduce a Global Prompt $\mathbf{P}_{Global}$, a learnable one-dimensional vector, to provide guidance for improving the overall and global exposure of the image. Specifically, in PIM, the Global Prompt is reshaped through repeating copies to match the size of the feature map $F$ from the Encoder, resulting in the Global Prompt Map ($\mathbf{M}_{Global} \in \mathbb{R}^{H' \times W' \times C}$). This design allows it to handle input images of arbitrary size. Notably, unlike the Local Prompt, which helps enhance local details, $\mathbf{M}_{Global}$ can adjust the overall tone of the image and improve exposure correction quality. Therefore, in this work, we employ both the Local Prompt and Global Prompt together to leverage their synergy, and the process within PIM can be expressed as follows:

$$\mathbf{PIM}(F) = \mathbf{LP\text{-}CA}(F, \mathbf{M}_{Local}) + \mathbf{M}_{Global}. \tag{7}$$

### 3.4 LOSS FUNCTIONS

To train CAGGLE, both reconstruction loss $\mathcal{L}_{recon}$ and color naming loss $\mathcal{L}_{cn}$ are utilized.

**Reconstruction loss** We utilize a reconstruction loss to minimize the discrepancy between the exposure correction result $I_{out}$ and the ground-truth image $I_{gt}$. The reconstruction loss, denoted as $\mathcal{L}_{recon}$, is calculated as the L1 distance between $I_{out}$ and $I_{gt}$ in the RGB color space as:

$$\mathcal{L}_{recon} = ||I_{out} - I_{gt}||_1. \tag{8}$$

**Color naming loss** To train the color estimation network $h$ in PIM, which predicts the weights associated with $\mathbf{P}_{Local}$, we employ the output of Color Naming model as the training target. This loss, referred to as the color naming loss $\mathcal{L}_{cn}$, is expressed as follows:

$$\mathcal{L}_{cn} = ||h(F)_\uparrow - I_{cn}||_2^2, \tag{9}$$

where $I_{cn}$ denotes the color probability map of the input image from Color Naming model, and $\uparrow$ indicates the bilinear upscaling operation to ensure dimension matching between $I_{cn}$ and the weight map from the color estimation network $h$.

Lastly, our final objective function $\mathcal{L}_{CAGGLE}$ to optimize the Encoder, PIM, and the Decoder is as follows:

$$\mathcal{L}_{CAGGLE} = \mathcal{L}_{recon} + \mathcal{L}_{cn}. \tag{10}$$

Table 1: Quantitative results on MSEC (Afifi et al., 2021), SICE (Cai et al., 2018) and LCDP (Wang et al., 2022) in terms of PSNR↑/SSIM↑. The best score is displayed in Red, the second in Blue.

| Method | #Params | MSEC | | | SICE | | | LCPD |
|---|---|---|---|---|---|---|---|---|
| | | Under | Over | Average | Under | Over | Average | Average |
| CLAHE (Zuiderveld, 1994) | - | 16.77/0.6211 | 14.45/0.5842 | 15.38/0.5990 | 12.69/0.5037 | 10.21/0.4878 | 11.45/0.4942 | 16.33/0.6420 |
| RetinexNet (Wei et al., 2018) | 0.840M | 12.13/0.6209 | 10.47/0.5953 | 11.14/0.6048 | 12.94/0.5171 | 12.87/0.5212 | 12.90/0.5212 | 19.25/0.7041 |
| ZeroDCE (Guo et al., 2020) | 0.079M | 14.55/0.5887 | 10.40/0.5142 | 12.06/0.5441 | 16.92/0.6330 | 7.11/0.4292 | 12.02/0.5311 | 12.59/0.6530 |
| RUAS (Liu et al., 2021) | 0.002M | 13.43/0.6807 | 6.39/0.4655 | 9.20/0.5515 | 16.63/0.5589 | 4.54/0.3196 | 10.59/0.4393 | 13.76/0.6060 |
| SCI (Ma et al., 2022) | 0.001M | 9.97/0.6681 | 5.84/0.5190 | 7.49/0.5786 | 17.86/0.6401 | 4.45/0.3629 | 12.49/0.5051 | 11.87/0.5234 |
| MSEC (Afifi et al., 2021) | 7.040M | 20.52/0.8129 | 19.79/0.8156 | 20.08/0.8210 | 19.62/0.6512 | 17.59/0.6560 | 18.58/0.6536 | 20.38/0.7800 |
| LCDPNet (Wang et al., 2022) | 0.960M | 22.35/0.8650 | 22.17/0.8476 | 22.30/0.8552 | 17.45/0.5622 | 17.04/0.6463 | 17.25/0.6043 | 23.24/0.8420 |
| ECLNet (Huang et al., 2022c) | 0.018M | 22.37/0.8566 | 22.70/0.8673 | 22.57/0.8631 | 22.05/0.6893 | 19.25/0.6872 | 20.65/0.6861 | 22.44/0.8061 |
| FECNet (Huang et al., 2022b) | 0.150M | 22.96/0.8598 | 23.22/0.8748 | 23.12/0.8688 | 22.01/0.6737 | 19.91/0.6961 | 20.96/0.6849 | 22.41/0.8402 |
| DRBN-ENC (Huang et al., 2022a) | 0.580M | 22.72/0.8544 | 22.11/0.8521 | 22.35/0.8530 | 21.89/0.7071 | 19.09/0.7229 | 20.49/0.7150 | 22.09/0.8271 |
| MSEC+DA (Wang et al., 2023b) | 7.040M | 21.53/0.8590 | 21.55/0.8750 | 21.54/0.8670 | 20.94/0.7546 | 17.49/0.6640 | 19.22/0.7093 | 21.05/0.8119 |
| ECLNet+ERL (Huang et al., 2023) | 0.018M | 22.90/0.8624 | 22.58/0.8676 | 22.70/0.8655 | 22.14/0.6908 | 19.47/0.6982 | 20.81/0.6945 | 22.63/0.8096 |
| PromptIR (Potlapalli et al., 2023) | 34.164M | 15.80/0.7391 | 16.73/0.7852 | 16.36/0.7668 | 22.51/0.6955 | 19.29/0.6849 | 20.90/0.6902 | 23.49/0.8513 |
| CSEC (Li et al., 2024) | 1.364M | 22.18/0.8502 | 22.69/0.8662 | 22.73/0.8638 | 20.79/0.7031 | 20.02/0.7093 | 20.41/0.7062 | 23.63/0.8550 |
| **CAGGLE** | 1.233M | 23.12/0.8660 | 23.31/0.8749 | 23.20/0.8695 | 24.18/0.7096 | 21.94/0.7462 | 23.06/0.7279 | 24.01/0.8647 |

Table 2: Color difference metrics, $\Delta E_{2000} \downarrow$ and $\Delta E_{ab} \downarrow$, defined in the CIELAB color space on the SICE (Cai et al., 2018) dataset. The best score in Red, the second in Blue.

| Metrics | ZeroDCE | RUAS | SCI | ECLNet | FECNet | DRBN-ENC | PromptIR | CSEC | **CAGGLE** |
|---|---|---|---|---|---|---|---|---|---|
| $\Delta E_{2000}\downarrow$ (Sharma et al., 2005) | 23.78 | 29.59 | 25.64 | 9.15 | 8.85 | 8.68 | 9.27 | 8.72 | 6.68 |
| $\Delta E_{ab}\downarrow$ (Sharma & Bala, 2017) | 29.31 | 37.05 | 31.22 | 11.58 | 11.19 | 11.02 | 11.59 | 11.39 | 8.54 |

## 4 EXPERIMENTS

### 4.1 IMPLEMENTATION DETAILS

For training, we adopt the Adam optimizer with a patch size of 256×256 and a batch size of 16. The total number of epochs is set to 500, and the learning rate is $2 \times 10^{-4}$. Additionally, we set $N = 6$ and $C = 128$ for Local Prompt. The implementation is based on the *PyTorch* framework, utilizing a single NVIDIA RTX 4090 GPU, and our code will be released upon acceptance.

**Datasets.** The training and benchmark settings follow the existing exposure correction tasks (Huang et al., 2022c;b; Li et al., 2024; Wang et al., 2022). We train the network on three multi-exposure datasets, including the Multiple Exposure (ME) (Afifi et al., 2021), Single Image Contrast Enhancement (SICE) (Cai et al., 2018), and LCDP (Wang et al., 2022). The ME dataset contains 17,675 training images, 750 validation images, and 5,905 test images across five exposure levels. The SICE dataset consists of sequences of 4–7 images of the same content at different exposure levels, and the LCDP dataset contains 1,700 different scenes with both over- and under-exposure to facilitate training and evaluation. In addition, to confirm the performance under different low-light conditions, we use RELLISUR (Aakerberg et al., 2021) dataset, which has multiple low-light exposure settings. This contains 850 distinct sequences with three different scales, and 2,550 pairs are provided, and we only use the ×1 scale for low-light image enhancement. Each image has exposure values ranging from -4.5 to -2.5 or -5.0 to -3.0 in 0.5 intervals, resulting in 12,750 paired images and RELLISUR dataset is split into 85% for training, 5% for validation, and 10% for testing.

**Comparative methods.** We compare CAGGLE to several state-of-the-art exposure correction methods, including CSEC (Li et al., 2024), LCPDNet (Wang et al., 2022), ECLNet (Huang et al., 2022c), FECNet (Huang et al., 2022b) and pluggable approaches DA (Wang et al., 2023b), ERL (Huang et al., 2023), and ENC (Huang et al., 2022a). Additionally, for low-light image enhancement, we benchmark CAGGLE against various previous methods, including Zero-DCE (Guo et al., 2020), Retinex-Net (Wei et al., 2018), RUAS (Liu et al., 2021), KinD (Zhang et al., 2019), GLADNet (Wang et al., 2018), MIRNet (Zamir et al., 2020), and MBLLEN (Lv et al., 2018). Each comparison model is evaluated using its official network parameter weights and reproduced with its official code for the RELLISUR (Aakerberg et al., 2021) dataset. For PromptIR (Potlapalli et al., 2023), we set the number of prompts equal to the number of exposure values in each dataset: 5 for MSEC, 2 for SICE, 2 for LCDP, and 5 for RELLISUR. The evaluations are measured in terms of Peak-Signal-to-Noise-Ratio (PSNR) and Structural Similarity (SSIM).

### 4.2 PERFORMANCE EVALUATION

We present the exposure correction results on the representative multi-exposure datasets MSEC, SICE, and LCDP in Table 1. On the MSEC dataset, our proposed method consistently outperforms previous approaches in terms of average PSNR and SSIM values. CAGGLE achieves the highest

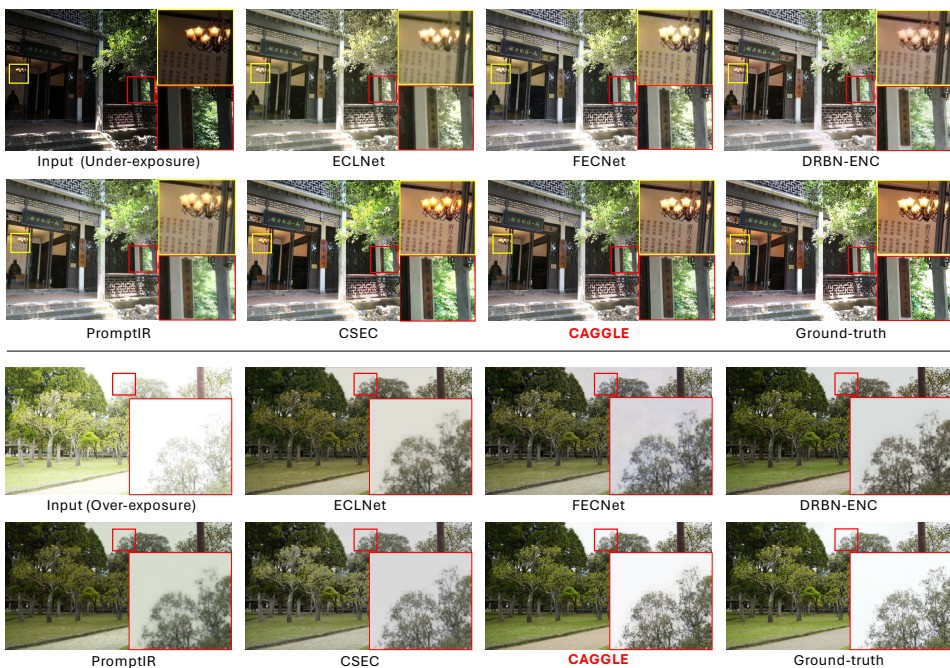

Figure 3: Qualitative comparisons on the SICE (Cai et al., 2018) dataset. **(Top)** Examples of images enhanced from under-exposed condition. **(Bottom)** Images enhanced from over-exposed condition. To facilitate precise quality comparison, zoomed-in details are provided for each case.

scores in all cases except one, where it ranks second in SSIM for the over-exposed case, trailing by only 0.001. Similarly, for the SICE dataset, our approach demonstrates the best performance except for the SSIM value in the under-exposed case, where it also ranks second. Compared to the previous SOTA methods (Huang et al., 2022b;a), our method achieves a large gain of 2.1 dB in PSNR and 0.0108 in SSIM. Lastly, on the LCDP dataset, our method achieves the highest results, outperforming CSEC (Li et al., 2024).

Additionally, to evaluate color correction performance, we conducted a comparison using $\Delta E_{2000}$ (Sharma et al., 2005) and $\Delta E_{ab}$ (Sharma & Bala, 2017) metrics in the LAB color space. Table 2 presents the results, demonstrating that our method also excels in color correction. CAGGLE shows a performance improvement of more than 2.0 in both $\Delta E_{2000}$ and $\Delta E_{ab}$ compared to previous methods. Achieving state-of-the-art performance across three distinct multi-exposure correction datasets demonstrates that our approach, by employing Global and Local Prompts, is highly effective for image exposure correction.

Fig. 3 shows the visual results of under- and over-exposure on the SICE dataset. For under-exposed images, CAGGLE restores brightness closer to the ground truth, balancing shadows without washing out highlights. In zoomed-in areas, CAGGLE preserves fine details, especially textures like writing on signs, and delivers the sharpest and most refined details compared to other methods. It accurately reproduces natural colors in greenery and restores details and color in over-exposed areas, such as tree leaves and sky, where other methods tend to brighten or wash out. Overall, CAGGLE maintains superior detail and natural colors, demonstrating both quantitative and qualitative superiority.

### 4.3 EXTENSION TO LOW-LIGHT ENHANCEMENT

Table 3 presents results on the RELLISUR dataset (Aakerberg et al., 2021), where CAGGLE outperformed MIRNet (Zamir et al., 2020) with a 0.24dB gain in PSNR and 0.01 in SSIM, despite having only 1.2M parameters compared to MIRNet's 31.8M. This highlights CAGGLE's potential for low-light enhancement as well as multi-exposure correction. Visual comparisons in Fig. 4 show CAGGLE closely matches the ground-truth brightness and excels in brick detail depiction.

### 4.4 ANALYSIS OF PROMPTS

To achieve consistent exposure correction results regardless of the input exposure level, it is important to extract features that are invariant to exposure variations. Our input-specific prompts in

Table 3: Quantitative results on the RELLISUR (Aakerberg et al., 2021) dataset. The best score is highlighted in **Red**, the second in **Blue**.

| Metrics | ZeroDCE | RetinexNet | RUAS | EnlightenGAN | KinD | GLADNet | MBLLEN | MIRNet | PromptIR | CSEC | **CAGGLE** |
|---------|---------|------------|------|--------------|------|---------|--------|--------|----------|------|--------|
| PSNR↑ | 12.99 | 15.43 | 11.92 | 11.61 | 15.84 | 21.09 | 17.52 | 21.62 | 20.77 | 10.66 | **21.86** |
| SSIM↑ | 0.44 | 0.34 | 0.34 | 0.39 | 0.49 | 0.69 | 0.60 | 0.77 | 0.75 | 0.30 | **0.78** |
| #Params (M) | 0.079 | 0.840 | 0.002 | 8.370 | 8.540 | 1.132 | 0.450 | 31.787 | 34.164 | 1.364 | 1.233 |

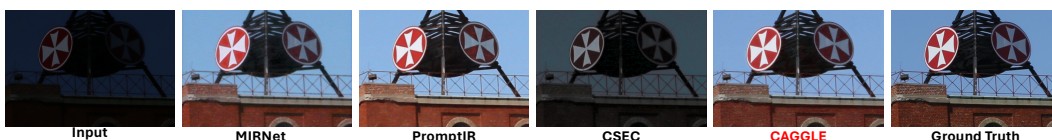

Figure 4: Visual comparison on the RELLISUR (Aakerberg et al., 2021) dataset. From *left* to *right*: MIRNet (Zamir et al., 2020), PromptIR (Potlapalli et al., 2023), CSEC (Li et al., 2024), CAGGLE (our approach), and the ground-truth image.

CAGGLE dynamically interact with deep features, transforming over-exposed and under-exposed features into distinct representations for enhanced exposure correction. To verify the impact of our prompts, we compare cosine similarity values between images of the same scene but with different exposures (*i.e.,* under-exposure and over-exposure) in the feature space, before and after the PIM. (please see Appendix A.2 for more details).

The visualization of cosine similarity in Fig.5 (a) shows low similarity values (blue) before applying PIM and high similarity values (red) afterward. These analyses demonstrate that CAGGLE enhances performance by maintaining feature consistency between over- and under-exposed images, in line with previous approaches (Huang et al., 2022a; 2023; Yao et al., 2023). To further illustrate the consistency of our method, we present improved results of state-of-the-art models on the same scene under different exposure levels in Fig. 5 (b)-(e). The zoomed-in region shows that CAGGLE better preserves fine details compared to other methods (Huang et al., 2022b; Li et al., 2024) and minimizes color differences caused by under- and over-exposure. Our method produces the most consistent outputs across varying exposures, outperforming (Huang et al., 2022b; Li et al., 2024) in both the quantitative measure of $\Delta E_{ab}$ and qualitative results.

## 4.5 ABLATION STUDY

**Impact of Prompts**  In PIM, we introduce two prompts: Local Prompt and Global Prompt. To analyze their impact, we present ablation experiments in Table 4. Case (a) in Table 4, which does not employ $\mathbf{M}_{Global}$ and $\mathbf{M}_{Local}$ (baseline), yields the lowest performance on average, while the case (b), applying only $\mathbf{M}_{Local}$, and case (c) applying only $\mathbf{M}_{Global}$, exhibit improvements in average PSNR and SSIM over the baseline. Notably, applying only $\mathbf{M}_{Local}$ results in a significant performance improvement on both under- and over-exposed images, whereas applying $\mathbf{M}_{Global}$ shows better results on over-exposed images. This suggests that Local Prompt excels at refining spatial and localized features necessary for exposure correction, while Global Prompt effectively handles natural tone adjustment for over-exposure. Lastly, case (d), which combines both $\mathbf{M}_{Local}$ and $\mathbf{M}_{Global}$, outperforms the other ablation cases, demonstrating that Global and Local Prompts create a synergistic effect to achieve significantly enhanced results.

Fig. 6 presents the visual results of the prompt ablation experiments, with the corresponding $\Delta E_{ab}$ values indicating the color correction performance. Employing only $\mathbf{M}_{Local}$, showcases enhanced color, spatial, and structural details, while using only $\mathbf{M}_{Global}$, achieves effective tonal adjustments,

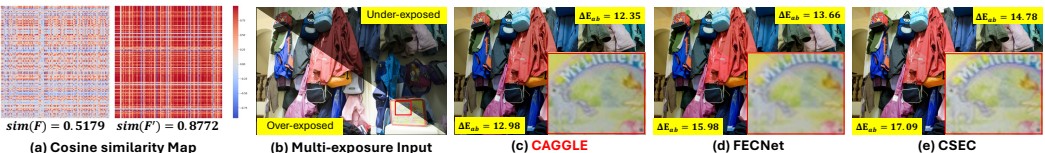

Figure 5: Visualization of prompt analysis. **(a)** Cosine similarity results between images of the same scene with different exposure values. The left image shows the similarity map before applying PIM, while the right one shows it after applying PIM. **(b)-(e)** Visual comparison for the same scene with different exposures with FECNet (Huang et al., 2022b) and CSEC (Li et al., 2024).

Table 4: Ablations on local and global prompts.

| Case | $M_{Local}$ | $M_{Global}$ | Under | | Over | | AVG. | |
|------|-------------|--------------|-------|------|------|------|------|------|
| | | | PSNR | SSIM | PSNR | SSIM | PSNR | SSIM |
| (a) | · | · | 23.28 | 0.7075 | 20.36 | 0.7215 | 21.82 | 0.7145 |
| (b) | ✓ | · | 23.57 | 0.7081 | 21.10 | 0.7262 | 22.34 | 0.7171 |
| (c) | · | ✓ | 23.19 | 0.6936 | 21.23 | 0.7380 | 22.21 | 0.7158 |
| (d) | ✓ | ✓ | 24.18 | 0.7096 | 21.94 | 0.7462 | 23.06 | 0.7279 |

Table 5: Ablations on color names constraint.

| Case | color names | PSNR↑ | SSIM↑ | $\Delta E_{2000}$↓ | $\Delta E_{ab}$↓ |
|------|-------------|-------|-------|-------------------|------------------|
| (a) | · | 22.31 | 0.7184 | 7.58 | 9.70 |
| (b) | hard-code | 22.61 | 0.7192 | 7.04 | 8.94 |
| (c) | trainable | 23.06 | 0.7279 | 6.68 | 8.54 |

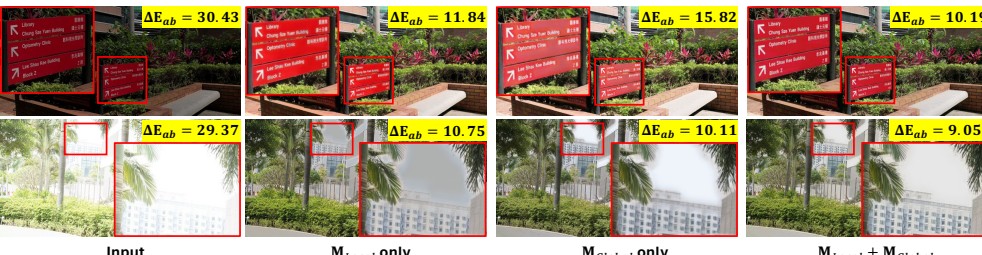

Figure 6: Visual results of using only $M_{Local}$, only $M_{Global}$, and both for the input image.

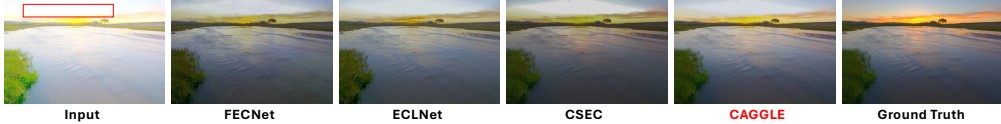

Figure 7: Visual results on over-saturated region. Although CAGGLE also struggles to correct this region, it produces a more natural result overall compared to other methods.

improving the background and structure more distinguishable. Finally, combining both global and local prompts captures the strengths of each approach and shows the best improvement over $\Delta E_{ab}$.

**Effectiveness of Color Naming Model** In Sec. 3.3.1, we introduce the color estimation network $h$ to predict the weights for $P_{Local}$, and to ensure that $P_{Local}$ provides color-aware information, we apply supervision using color names to train $h$ (Sec. 3.3.1). To validate this, we compared the results on the SICE (Cai et al., 2018) dataset as shown in Table 5. Case (a) represents the scenario where the Color Naming model is not used in the color estimation network (w/o $\mathcal{L}_{cn}$). Case (b) employs the probability map of the 6 color names estimated by the Color Naming model (Van De Weijer et al., 2009) directly as the weight for $P_{Local}$, without employing a color estimation network, while case (c) represents CAGGLE approach. Case (a) already achieves state-of-the-art performance compared to existing methods, while case (b) shows improvements in PSNR, SSIM, $\Delta E_{2000}$, and $\Delta E_{ab}$ over case (a). Additionally, our proposed approach, case (c), further enhances all metrics. This demonstrates that our method effectively integrates color names into the prompts, resulting in color-aware prompts that improve overall performance.

## 5 LIMITATIONS

While the prompts learn and provide useful information in the feature space for image enhancement, improvement remains challenging in extreme cases where the input image lacks sufficient information, such as over-saturated regions where pixel values are close to 255 (as shown in the red box of Fig. 7). To address this issue, we are exploring the application of generative models in areas that contain missing information. We will prioritize and continue to solve this problem in future work.

## 6 CONCLUSIONS

In this work, we tackle exposure correction through color-aware prompt learning with both Global and Local Prompts. We emphasize spatially-aware adjustment and introduce a novel color-aware prompt design incorporating color names. Our method, **CAGGLE**, enhances local details like color and structure through the Local Prompt while managing overall tone via the Global Prompt. Additionally, we propose **LP-CA**, which enhances Local Prompt performance. By leveraging these prompt techniques, CAGGLE achieves state-of-the-art results on multi-exposure benchmarks, effectively balancing global tone adjustment with local detail enhancement.

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

# A APPENDIX

## A.1 ENCODER AND DECODER

As described in Fig. 2, we utilize simple U-shaped network as our CAGGLE backbone. Table 6 presents the detailed architecture of the Encoder and Decoder. In Table 6, the Conv-block consists of a convolution operation with a stride of 1 and padding of 1.

Table 6: Specification of the CAGGLE backone architecture.

| Stage | Operations | Outputs |
|---|---|---|
| Enc-1 | Conv-block, $3 \times 3$
Conv-block, $3 \times 3$
batchnorm2d(32) | $h \times w \times 32$
$h \times w \times 32$
$h \times w \times 32$ |
| Enc-2 | Conv-block, $3 \times 3$
PixelShuffle(2)
Conv-block, $3 \times 3$
batchnorm2d(64) | $h \times w \times 16$
$h/2 \times w/2 \times 64$
$h/2 \times w/2 \times 64$
$h/2 \times w/2 \times 64$ |
| Enc-3 | Conv-block, $3 \times 3$
PixelShuffle(2)
Conv-block, $3 \times 3$
batchnorm2d(128) | $h/2 \times w/2 \times 32$
$h/4 \times w/4 \times 128$
$h/4 \times w/4 \times 128$
$h/4 \times w/4 \times 128$ |
| PIM | Prompt Interaction module | $h/4 \times w/4 \times 128$ |
| Dec-1 | Conv-block, $3 \times 3$
PixelUnshuffle(2) | $h/4 \times w/4 \times 256$
$h/2 \times w/2 \times 64$ |
| - | *Skip-connection* with Enc-2 | $h/2 \times w/2 \times 128$ |
| Dec-2 | Conv-block, $3 \times 3$
PixelUnshuffle(2)
Conv-block, $3 \times 3$
Conv-block, $3 \times 3$ | $h/2 \times w/2 \times 128$
$h \times w \times 32$
$h \times w \times 32$
$h \times w \times 32$ |
| - | *Skip-connection* with Enc-1 | $h \times w \times 64$ |
| Dec-3 | Conv-block, $3 \times 3$
Conv-block, $3 \times 3$
Conv-block, $1 \times 1$ | $h \times w \times 32$
$h \times w \times 32$
$h \times w \times 3$ |

## A.2 COSINE SIMILARITY

In Sec. 4.4 of our main manuscript, to assess the similarity between the feature representations of under- and over-exposed images, we employed cosine similarity as a metric. Cosine similarity provides a measure of alignment between feature vectors, with values ranging from -1 to 1, where higher values indicate greater similarity. The formula for cosine similarity used is as follows:

$$sim(Fea^U, Fea^O) = \frac{Fea^U \cdot Fea^O}{||Fea^U||_2 \cdot ||Fea^O||_2}, \tag{11}$$

where $Fea^U$ and $Fea^O$ represent the features of under-exposure and over-exposure, respectively.

Additionally, the following process is carried out to generate the cosine similarity map shown in Fig. 5 **(a)** and **(b)** of the main manuscript.

1. To facilitate calculation, the input images is resized to $256 \times 256$.

2. Feature Extraction: Two feature maps are extracted from the corresponding layers of the network, one for the under-exposed condition and another for the over-exposed condition. For Encoder and PIM, each feature map has spatial dimensions of $64 \times 64$ with 128 channels.

3. Flattening the Spatial Dimensions: To enable pairwise comparison, we first flatten the spatial dimensions of the feature maps. Each feature map is reshaped from a $64 \times 64$

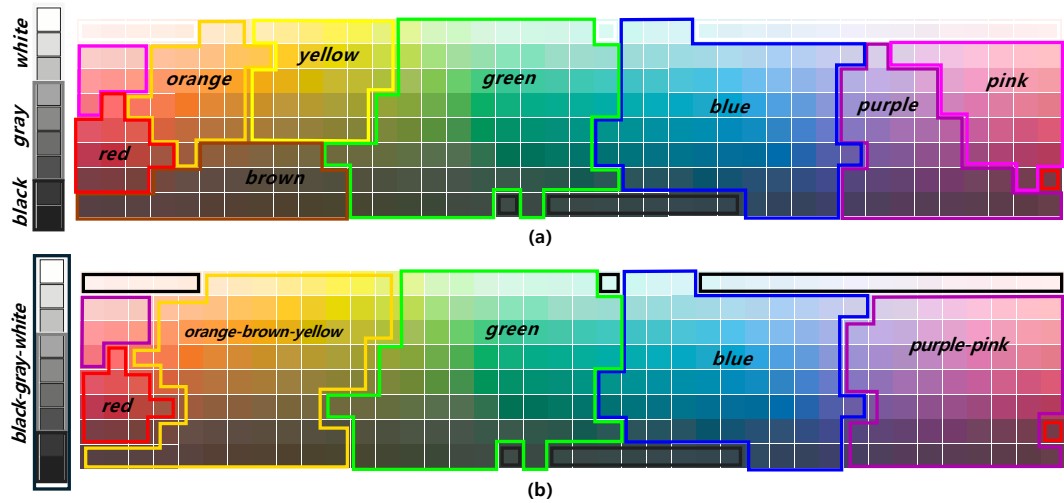

Figure 8: (a) 11 color names from Van De Weijer et al. (2009) organized in the Munsell color chart. (b) 6 color names from Serrano-Lozano et al. (2024) organized in the Munsell color chart.

spatial grid into a vector of size 4,096, resulting in a flattened feature matrix of shape [128, 4,096].

4. Normalization: We normalize the feature vectors along each channel to ensure that the cosine similarity measure is not influenced by the magnitude of the vectors. Each feature vector is normalized using the L2 norm.

5. Cosine Similarity Calculation: Cosine similarity is then computed between the flattened and normalized feature maps of the under-exposed and over-exposed images. This results in a $128 \times 128$ cosine similarity matrix, where each entry represents the similarity between the corresponding channels of the two feature maps.

6. Visualization: The computed cosine similarity matrix is visualized as a heatmap. The heatmap provides an intuitive view of how closely aligned the features are between the under-exposed and over-exposed images. The cosine similarity matrix is color-coded, where higher values indicate stronger similarity.

## A.3 COLOR NAME

A color term or color name refers to a word or phrase that represents a specific color, and the terms we use are based on the theory introduced by Berlin and Kay in *Basic Color Terms* (Berlin & Kay, 1991). Berlin and Kay argued that color perception is more influenced by physiological and perceptual factors than by cultural ones. They analyzed data collected from speakers of 20 languages across various language families and identified 11 basic color categories: *white, black, red, green, yellow, blue, brown, purple, pink, orange*, and *gray*. Since these color categories are based on human physiological processes, models that classify colors using these names are perceptually grounded.

Based on this theory, Van De Weijer et al. introduce a Color Naming model that categorizes the color of each part of real-world images. The Color Naming model does not aim to improve the naming of color patches, but instead focuses on accurately naming colors in real-world applications. In real-world scenarios, images are captured under various conditions such as different illuminations, reflections, unknown cameras, colored shadows, compression artifacts, acquisition aberrations, and unknown camera settings. Therefore, robust color naming is crucial for applied research. The Color Naming model uses *pLSA* (probabilistic Latent Semantic Analysis) (Hofmann et al., 1999) to model the probability of each pixel in an image belonging to a specific color name. As the objective of our method is to robustly recognize color names even under varying exposure levels and use this in the Local Prompt learning process, this approach aligns well with our research.

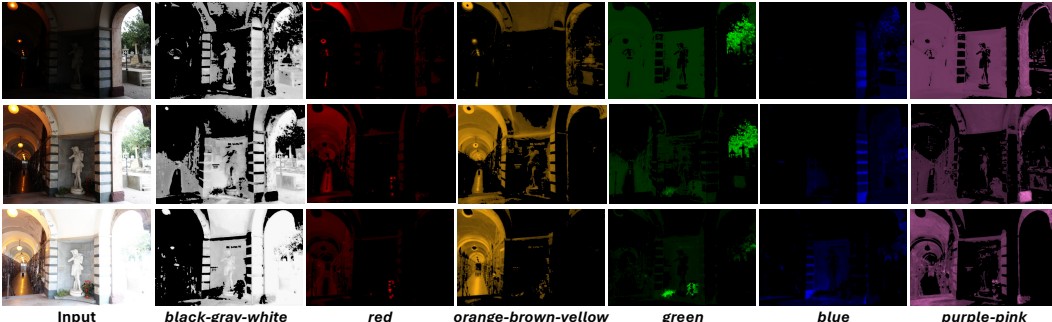

| Input | black-gray-white | red | orange-brown-yellow | green | blue | purple-pink |

Figure 9: Visualization of color probability maps generated by the Color Naming model for images of the same scene under different exposure levels.

| Input | black-gray-white | red | orange-brown-yellow | green | blue | purple-pink |

Figure 10: Visualization of color probability maps generated by the color estimation network ($h$) for images of the same scene under different exposure levels.

Meanwhile, Serrano-Lozano et al. grouped the 11 color names into 6 categories based on having the same hue and being implemented with changes in intensity. It is argued that grouping color names by hue is a more efficient approach. Similarly, our method uses 6 color names for training the network $h$.

To facilitate comprehension, we present the 11 color names from *Basic Color Terms*, along with the 6 color names employed by our method, in a Munsell color chart in Fig 8. Additionally, Fig.9 illustrates the visual outcomes of the probability maps for each color name under diverse exposure conditions for the same scene. Fig.10 also presents the visual outcomes of the probability maps for each color name generated by our color estimation network h. The color estimation network in CAGGLE produces results with fewer artifacts compared to the color naming map.

### A.3.1 ABLATIONS ON THE NUMBER OF COLOR NAMES

To evaluate the performance differences based on the number of color names, we conduct a comparative experiment in Table 7, applying (a) commonly used RGB categories, (b) the 6 color names we used, and (c) the 11 color names defined in *Basic Color Terms* to train $h$ for Local Prompt. The number of Local Prompt vectors is set to 3, 6, and 11, corresponding to the number of color names, and the results show that using color names (Table 7, (b), and (c)) outperformed using only RGB categories (Table 7,(a)), demonstrating the importance of using color names defined in Berlin & Kay (1991). Although (b) and (c) in Table 7 showed different superiority depending on the condition, on average, the 6 color names we used achieved the highest PSNR.

### A.4 PROMPT INTERACTION MODULE ON DECODER BRANCH

In here, we apply the Prompt Interaction Module (PIM) to the deep features between the *Encoder* and *Decoder*. This aligns with our intent of enhancing features before entering the decoder, similar to a normalization process. We study the result of applying PIM at each decoder layer, as shown in

Table 7: Ablation studies on the number of color names: **(a)** divides colors into RGB, **(b)** uses our proposed method, and **(c)** employs color names based on Van De Weijer et al. (2009).

| Case | color names | Under | | Over | | Average | |
|------|-------------|-------|------|------|------|---------|------|
| | | PSNR | SSIM | PSNR | SSIM | PSNR | SSIM |
| **(a)** | 3 | 22.90 | 0.7016 | 20.10 | 0.7026 | 21.50 | 0.7021 |
| **(b)** | **6** | **24.18** | 0.7096 | **21.94** | 0.7462 | **23.06** | 0.7279 |
| **(c)** | 11 | 23.81 | **0.7125** | 21.73 | **0.7471** | 22.77 | **0.7298** |

Table 8: Effect of prompt addition in each decoder stage. ✓represents the inclusion of the Prompt Interaction Module (PIM) before each decoder stage.

| Case | Dec-1 | Dec-2 | Dec-3 | Size (MB) | Under | | Over | | AVG. | |
|------|-------|-------|-------|-----------|-------|------|------|------|------|------|
| | | | | | PSNR | SSIM | PSNR | SSIM | PSNR | SSIM |
| backbone | · | · | · | 2.8 | 23.28 | 0.7075 | 20.36 | 0.7215 | 21.82 | 0.7145 |
| **(a)** | ✓ | · | · | **4.7** | **24.18** | **0.7096** | **21.94** | **0.7462** | **23.06** | **0.7279** |
| **(b)** | ✓ | ✓ | · | 5.2 | 23.55 | 0.7049 | 21.78 | **0.7433** | 22.66 | 0.7241 |
| **(c)** | ✓ | ✓ | ✓ | 5.5 | **23.83** | **0.7131** | **21.83** | 0.7401 | **22.83** | **0.7250** |

Table 8. While there were performance improvements in all cases (Table 8) (a), (b), and (c)), Table 8 (a) demonstrate the best results in terms of PSNR/SSIM, with the smallest network size.

## A.5 QUALITATIVE RESULTS

We present more qualitative results from LCDP (Wang et al., 2022).

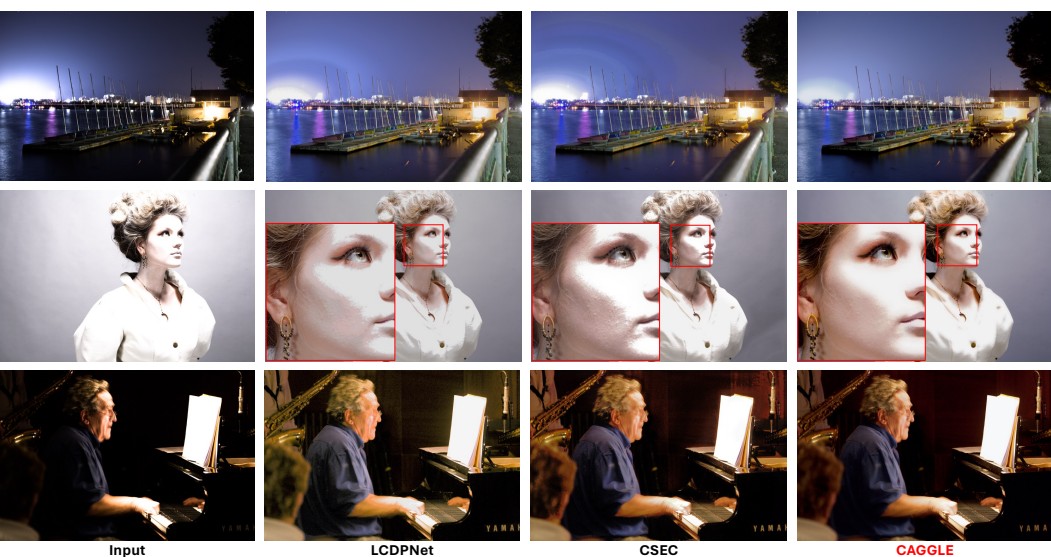

Figure 11: Visual examples on multi-exposure images.

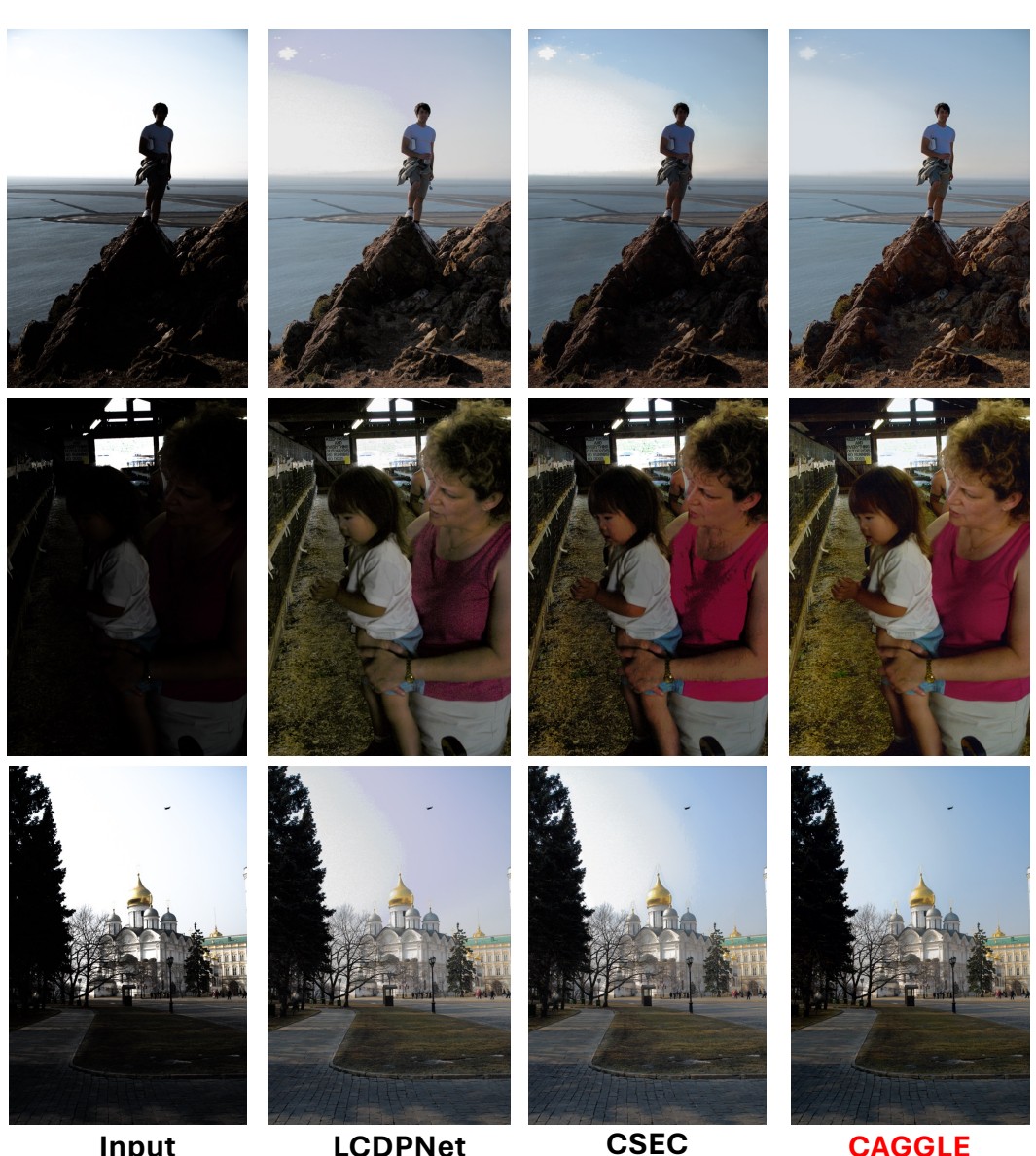

Figure 12: Visual examples on multi-exposure images.

