# OpenReview forum: "CAGGLE: Color-Aware Guidance with Global and Local Prompts for Exposure Correction"
_ICLR.cc/2025/Conference — ICLR 2025 Conference Withdrawn Submission_

### Official Review · Reviewer_YUBW · 2024-10-21

**Soundness:** 3
**Presentation:** 3
**Contribution:** 3
**Rating:** 6
**Confidence:** 5

**Summary:**

This paper proposes a method to correct images with under- and over-exposure problems by introducing the color-aware guidance via prompt-based learning. A prompt interaction module (PIM) is proposed, which leverages a global prompt for adjusting color tones and a local prompt for maintaining color consistency. The proposed method is evaluated on three public datasets where the results show its effectiveness against existing methods.

***I am leaning to accept this paper due to the proposed method is technically sound and achieves good results.***

**Strengths:**

+ There are indeed many images suffer from mixed exposure problems. It is interesting to explore the prompt learning for correcting such images by utilizing defined color names.

+ The effectiveness of global and local prompt learning for image exposure correction has been justified with both quantitative and visual results.

**Weaknesses:**

***The introduction and Figure 1 do not motivate this work well.***
The Abstract emphasizes that existing methods may fail in scenes with both under- and over-exposure problems and in over-saturated regions. However, the first image shown in Figure 1 is under-exposed and the other image is over-exposed. While they do not contain obvious mixed exposure problems, the results between the existing methods and the proposed one do not show significant differences that can indicate the problems suffered by existing methods. This results in a weak motivation for introducing color-aware guidance for image correction. Including examples in Figure 1 that better demonstrate the mixed exposure and over-saturation issues highlighted in the abstract would strengthen the paper's motivation.

***Missing Important Results.***
While this paper relies on the Color Naming model (Serrano-Lozano et al., 2024) to provide intermediate supervision, the comparisons to this method are necessary (but are missing) as it is an image enhancement method.

The claimed local prompt map conditioned Cross-Attention (LP-CA) is claimed to be novel but there is no justification for its novelty and the corresponding ablation results.



***A Minor Issue.***
It would be better to add references for the methods in Figure 1. Otherwise, it makes it hard to identify those works.

**Questions:**

Questions&Suggestions.
+ Please add comparisons to (Serrano-Lozano et al., 2024) and ablation results about LP-CA. Please explicitly clarify why LP-CA represents significant novel designs.

+ It would be great to add visualizations of features before and after processed by PIM (i.e., F and F’).

+ It would be good to discuss and summarize the relations/differences between the proposed method and existing methods in each subsection of Related Works.

+ There are three related works [A,B,C] not discussed. While [A] leverages intermediate HDR transformation to learn image details for under/over-exposure correction, [B] learns a set of post-processing operations for over/under-exposure correction. [C] utilizes prompt learning to enhance backlit images. Although [C] is for a slightly different task, discussing and comparing it would help appreciate the originality of the proposed work.

[A] Image Correction via Deep Reciprocating HDR Transformation, CVPR 2018
[B] Exposure: A White-Box Photo Post-Processing Framework, TOG 2018
[C] Iterative Prompt Learning for Unsupervised Backlit Image Enhancement, ICCV 2023

+ Figure 2 is not self-contained and the data/feature flow is hard to trace. For example, it is hard to understand where P_Local and P_Global come from and how they are transformed into M_Local and M_Global. It is not clear which Color Naming Model is used here.

+ It is not clear why the authors chose to evaluate the low-light image enhancement on a super-resolution dataset (Aakerberg et al., 2021). Meanwhile, such comparisons may not be meaningful as those low-light enhancement methods are not state-of-the-art. Instead, I would suggest the authors focus on the exposure correction task.

---

### Official Review · Reviewer_L4xU · 2024-10-22

**Soundness:** 3
**Presentation:** 2
**Contribution:** 3
**Rating:** 5
**Confidence:** 5

**Summary:**

This paper proposes a color-aware guidance approach for exposure correction, employing a global prompt for tone adjustment and a local prompt to maintain color consistency. The core idea is to use explicit color definition priors to guide color consistency in the image after exposure correction. Experimental results and ablation studies demonstrate the effectiveness of the proposed method.

**Strengths:**

1. This paper demonstrates that introducing explicit color definition priors from the Color Naming model ensures color consistency during exposure correction.
2.  The proposed method achieves state-of-the-art performance both qualitatively and quantitively, particularly in terms of color consistency in the visualized results.

**Weaknesses:**

1. The paper contains many subjective statements with unclear references. In the introduction (page 1, lines 38-41), the authors mention limitations of early research, but it is unclear which specific early research is being referred to and what limitations they had. In the method section, the organization is confusing, making the overall pipeline and the proposed modules difficult to read and follow.
2. There are some concerns regarding the proposed CAGGLE. Why does the paper prioritize learning the local prompt before the global prompt? Intuitively, global illumination information should be easier to learn. How would the performance change if the model learned the global prompt first, followed by the local prompt? Additionally, the Color Naming Loss computes loss based on the input color priors. However, based on the qualitative results, after exposure correction, the color of the ground truth may change, and different exposure correction levels result in varying color shifts. The reviewer is curious about how color consistency is ensured after exposure correction.
3. The computational cost of the proposed method, such as parameters, FLOPs, and runtime, should be reported for comparison.

**Questions:**

If the authors address the issues mentioned in the weaknesses, the reviewer would consider raising the score.

---

### Official Review · Reviewer_gUSo · 2024-10-30

**Soundness:** 2
**Presentation:** 3
**Contribution:** 3
**Rating:** 3
**Confidence:** 4

**Summary:**

This paper proposed a prompt-based exposure correction method, which incorporates the local and global prompt to enhance image features, and intergrates the naming model to guide the prompt learning. The novel Prompt Interaction Module (PIM) that seamlessly integrates the global and local prompts with the input image features.

**Strengths:**

*The authors claim that the proposed method is a new standard in exposure correction, leveraging prompt-based learning for improved color and exposure adjustments.

**Weaknesses:**

*The introduction of prompt learning to the exposure correction is commendable, but the explanation of local and global prompts is not very convincing, and the promotion effect of the color naming module on local prompt features has not been well explained.

*The explanation of the proposed method is not clear enough. The color naming module relies on existing methods, but there is a lack of clear explanation on how the local and global prompt modules are implemented.

**Questions:**

*How are local and global modules specifically implemented? Is it directly using existing methods? Is there any special design?

---

### Official Review · Reviewer_GNn2 · 2024-11-03

**Soundness:** 3
**Presentation:** 2
**Contribution:** 3
**Rating:** 6
**Confidence:** 4

**Summary:**

The authors propose a method to address the problem of exposure correction in images where both under-exposure and over-exposure occur within the same image. The proposed method introduces prompts to maintain global features and local features, aiming to solve this problem. In this process, they introduced a color naming model to enable stable learning of the proposed model. The authors not only demonstrate superiority over other existing methods in many benchmarks but also show the excellence of methods including learnable prompts such as PIM through various additional experiments, including ablation studies to verify the effectiveness of the Color Naming Model.

**Strengths:**

* It's very impressive that the authors attempted to solve a kined of information restoration problem that occurs despite selecting an appropriate exposure value to capture a scene with extreme brightness differences between areas. it is also a very practical issue. This is closely related to the area that inverse tone mapping in the HDR field aims to address.

* They introduced a color naming model to guide prompt learning, ensuring color consistency. I believe this is a technically valid approach that can align the latent space of the trained neural network model with the color space perceived by humans.

* This paper explains the proposed method in a detailed yet easy-to-understand manner.

**Weaknesses:**

* The proposed model lacks novelty as it is structurally similar to other common models.
* The authors need to visualize the latent space practically.
  * For example, 1) They can verify whether the encoder's output for a given image, when rearranged into pixel-level features, can be well-divided into K large groups by using K-means algorithm. Or 2) For the given dataset, the authors can extract pixel-level features, classify each pixel as over-exposed, under-exposed, or normally exposed, and then visualize these classifications using t-SNE. If this process is too difficult, they should at least perform a similar process at the image level.
  * Particularly, regarding Table 5 shown by the authors, there is a need to analyze how different latent spaces the models have from each other.

**Questions:**

* Rather than in limited environments like CCTV surveillance, I believe this algorithm could have an advantage over others in natural scenes that include both completely dark areas and intense light sources (like the top row of Figure 1). Is this perspective correct? If so, I would like to see this content included in the introduction.

* The authors need to visualize the latent space practically. For example, they should show whether the encoder's output for a given image, when rearranged into pixel-level features, can be well-divided into N large groups

* Let's assume we are given an image and N local prompts. An analysis that measures the IOU between a binary mask created by collecting pixels highly correlated with one local prompt and the actual pixel mask divided by the color naming model could be very interesting.

* Could you provide more details about the rationale behind using pixel-shuffle and un-shuffle layers?

* Some information is missing  or reproducibility. This includes the values for the beta terms of the Adam optimizer, etc.

---

### Note · Authors · 2024-11-15

**Comment:**

First of all, we are truly thankful for reviewers' comments.

Although valuable comments we provided from reviewers, we found there are much more works to compensate for our research.

Therefore, we chose to withdraw this paper.

Thank you.

**Withdrawal Confirmation:**

I have read and agree with the venue's withdrawal policy on behalf of myself and my co-authors.